# Molecular mechanism underlying desensitization of the proton-activated chloride channel PAC

**James Osei-Owusu[1†], Zheng Ruan[2†], Ljubica Mihaljević[1], Daniel S Matasic[3], Kevin Hong Chen[1], Wei Lü[2]\*, Zhaozhu Qiu[1,4]\***

[1]Department of Physiology, Johns Hopkins University School of Medicine, Baltimore, United States; [2]Department of Structural Biology, Van Andel Institute, Grand Rapids, United States; [3]Department of Medicine, Johns Hopkins University School of Medicine, Baltimore, United States; [4]Solomon H. Snyder Department of Neuroscience, Johns Hopkins University School of Medicine, Baltimore, United States

**\*For correspondence:**
wei.lu@vai.org (WL);
zhaozhu@jhmi.edu (ZQ)

[†]These authors contributed equally to this work

**Competing interest:** The authors declare that no competing interests exist.

**Abstract** Desensitization is a common property of membrane receptors, including ion channels. The newly identified proton-activated chloride (PAC) channel plays an important role in regulating the pH and size of organelles in the endocytic pathway, and is also involved in acid-induced cell death. However, how the PAC channel desensitizes is largely unknown. Here, we show by patch-clamp electrophysiological studies that PAC (also known as TMEM206/ASOR) undergoes pH-dependent desensitization upon prolonged acid exposure. Through structure-guided and comprehensive mutagenesis, we identified several residues critical for PAC desensitization, including histidine (H) 98, glutamic acid (E) 94, and aspartic acid (D) 91 at the extracellular extension of the transmembrane helix 1 (TM1), as well as E107, D109, and E250 at the extracellular domain (ECD)–transmembrane domain (TMD) interface. Structural analysis and molecular dynamic simulations revealed extensive interactions between residues at the TM1 extension and those at the ECD–TMD interface. These interactions likely facilitate PAC desensitization by stabilizing the desensitized conformation of TM1, which undergoes a characteristic rotational movement from the resting and activated states to the desensitized state. Our studies establish a new paradigm of channel desensitization in this ubiquitously expressed ion channel and pave the way for future investigation of its relevance in cellular physiology and disease.

## Editor's evaluation

This important study addresses the molecular mechanisms of the proton-activated chloride channel (PAC), a widely expressed ion channel involved in organelle pH homeostasis and acid-induced cell death. Convincing data based on structure-guided mutagenesis and molecular dynamics simulations provides new insight into the mechanism underlying channel desensitization under sustained acidic stimulation. The results are of interest to ion channel physiologists.

## Introduction

Opened by extracellular acidic pH, the proton-activated chloride (PAC) channel activity has been recorded by patch-clamp electrophysiology on the plasma membrane of a variety of mammalian cells for decades (*Auzanneau et al., 2003*; *Capurro et al., 2015*; *Kittl et al., 2020*; *Lambert and Oberwinkler, 2005*; *Ma et al., 2008*; *Sato-Numata et al., 2013*; *Yamamoto and Ehara, 2006*). Its molecular identity remained a mystery until recently when we and others identified *PACC1* (also known as

*TMEM206*) encoding the PAC channel through unbiased RNA interference screens (*Ullrich et al., 2019*; *Yang et al., 2019*). Under pathological conditions, the PAC channel contributes to acid-induced cell death and tissue damage (*Osei-Owusu et al., 2020*; *Ullrich et al., 2019*; *Wang et al., 2007*; *Yang et al., 2019*). The discovery of PAC's physiological function has been made possible by its molecular identification and subsequent localization study (*Osei-Owusu et al., 2021*; *Ullrich et al., 2019*; *Yang et al., 2019*). The PAC channel traffics from the cell surface to several intracellular organelles in the endocytic pathway, where it mediates pH-dependent chloride (Cl⁻) transport and serves in important housekeeping roles, including regulation of endosome acidification and macropinosome shrinkage (*Osei-Owusu et al., 2021*; *Zeziulia et al., 2022*).

With prolonged (electrical, chemical, or mechanical) stimulation, many ion channels undergo inactivation or desensitization as a potential mechanism to limit signal transduction. For example, sodium-selective acid-sensing ion channels (ASICs) often desensitize rapidly (in seconds) after proton-induced activation, thus producing characteristic transient currents (*Sutherland et al., 2001*; *Yoder et al., 2018*; *Zhang and Canessa, 2002*). As a new family of ion channels sensitive to acid, much attention has been focused on the structural and functional characterization of PAC's pH-sensing and gating mechanism. However, the desensitization profile of the PAC channel and its underlying molecular mechanism remain unclear. Notably, in addition to a high-pH resting closed state and a low-pH activated open state, cryo-electron microscopy (cryo-EM) studies revealed a third low-pH non-conducting state of the trimeric human PAC channel. This structure represents the dominant population at pH 4.0 and is also observed at pH 4.5. It has been proposed to be a pre-open or desensitized state of PAC (*Wang et al., 2022*; *Ruan et al., 2020*). However, the physiological relevance of this conformation is incompletely understood. Here, we performed electrophysiological studies and found that the PAC channel displays pH-dependent desensitization below pH 4.6 with increased magnitude and faster kinetics at more acidic conditions. We further identified several key residues regulating PAC channel desensitization, which are concentrated at the interface between the extracellular domain (ECD) and the transmembrane domains (TMD). Combined with structural analysis and molecular dynamics (MD) simulations, our study reveals a distinct desensitization mechanism for the human PAC channel characterized by a dramatic rotational movement of transmembrane helix 1 (TM1).

## Results

### The PAC channel exhibits pH-dependent desensitization

We have previously noted that the PAC currents desensitize over time when exposed to acidic pH below 4.6 (*Ruan et al., 2020*). To systematically characterize the desensitization profile of the PAC channel, we performed whole-cell patch-clamp recordings and measured the endogenous proton-activated Cl⁻ currents in HEK293T cells. To activate PAC, we perfused extracellular acidic solutions ranging from pH 5.0 (half-maximal activation or pH$_{50}$ of PAC) to 3.6. While the PAC currents at pH 5.0 were stable during the course of the measurement, the channel showed progressively more obvious desensitization with increasingly acidic stimuli (*Figure 1A*). To quantify the current desensitization, we normalized the decayed current after 30 s of exposure to acidic pH to the initial maximal current (*Figure 1B*). This analysis revealed that the desensitization of the PAC currents is pH-dependent, that is, the lower the pH, the greater the degree of decay (*Figure 1A and B*). We fitted the current decay at pH 3.6 using a single exponential function and calculated the desensitization time constant of 64.1±6.9 s (mean ± SEM, n=6). The peak current densities at pH 4.6–3.6, however, were similar (*Figure 1C*), indicating that the PAC channel completely recovers from the previous desensitization during neutral pH treatment between two acidic stimuli. The recovery happened in few seconds after the earlier channel desensitization at pH 4.0 (*Figure 1—figure supplement 1A, B*). A similar pH-dependent desensitization profile was also observed for transiently overexpressed human PAC cDNA in *PAC* knockout (KO) HEK293T cells, which generated much larger channel currents (*Figure 1D–F*). Interestingly, its degree and kinetics (time constant: 27.1±4.7 s [mean ± SEM, n=7] at pH 3.6) were greater and faster compared to the endogenous currents at the same pH. This difference suggests a potential endogenous factor/mechanism limiting PAC desensitization that is overwhelmed by the large number of overexpressed channels. For the rest of the study, we performed the structure–function analysis on PAC overexpressing-mediated currents in the KO cells, so we can assay various PAC mutants.

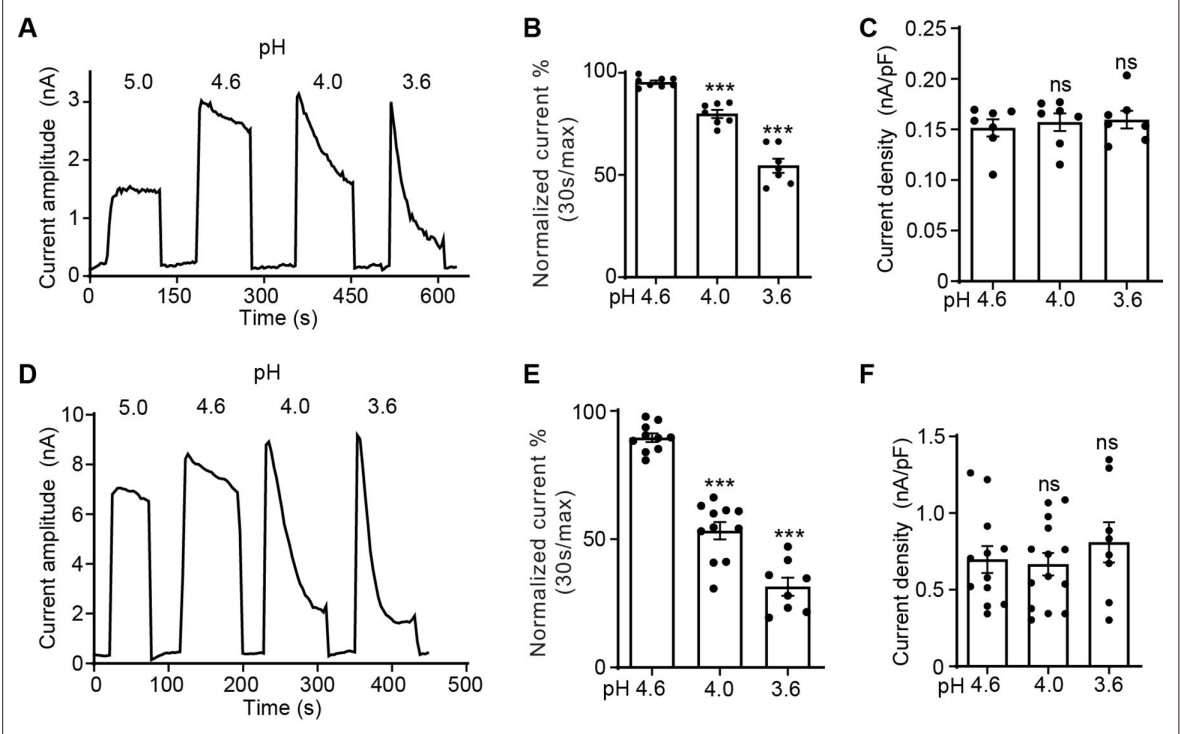

**Figure 1.** The proton-activated chloride (PAC) channel desensitizes at low pH. (**A**) Representative whole-cell current traces of the endogenous PAC current ($I_{Cl, H}$) in wild-type (WT) HEK293T cells induced by extracellular acidification at pH 5.0, 4.6, 4.0, and 3.6. Cells were maintained in pH 7.3 solution before acidic solution application. Currents were monitored by voltage-ramp protocol and traces shown are at +100 mV. (**B**) Percentage desensitization (mean ± SEM) of $I_{Cl, H}$ in WT HEK293T cells at pH 4.6, 4.0, and 3.6. Desensitized current after 30 s of acidic exposure was normalized to the initial maximum current of that recording and expressed as a percentage. ***p<0.001; one-way analysis of variance (ANOVA) with Bonferroni post hoc test. (**C**) Current densities (mean ± SEM) of the $I_{Cl, H}$ in WT HEK293T cells induced by acidic solutions pH 4.6, 4.0, and 3.6 at +100 mV. ns, not significant; one-way ANOVA with Bonferroni post hoc test. (**D**) Representative whole-cell current trace of PAC overexpressing current in *PAC* KO HEK293T cells transfected with human PAC cDNA. Currents at different pH values were monitored by voltage-ramp protocol and traces shown are at +100 mV. (**E**) Percentage desensitization (mean ± SEM) of PAC overexpressing current in *PAC* KO HEK293T cells at pH 4.6, 4.0, and 3.6. Desensitized current after 30 s of acidic exposure was normalized to the initial maximum current of that recording and expressed as a percentage. ***p<0.001; one-way ANOVA with Bonferroni post hoc test. (**F**) Current densities (mean ± SEM) of PAC overexpressing current induced by pH 4.6, 4.0, and 3.6 solutions at +100 mV. ns, not significant; one-way ANOVA with Bonferroni post hoc test.

The online version of this article includes the following source data and figure supplement(s) for figure 1:

**Source data 1.** Raw data for *Figure 1A–F*.

**Figure supplement 1.** The proton-activated chloride (PAC) channel does not exhibit steady-state desensitization and shows fast recovery from desensitization.

**Figure supplement 1—source data 1.** Raw data for *Figure 1—figure supplement 1A-D*.

The characteristics of PAC channel desensitization is quite different from that of ASICs. Compared to ASICs, PAC desensitizes much slower with a time constant of minutes rather than seconds (*Rook et al., 2020*; *Sutherland et al., 2001*; *Zhang and Canessa, 2002*). Even after minutes of low pH exposure, there were still large residual PAC currents. In addition, pre-treatment of mildly acidic pH (not low enough to induce substantial channel activation) did not reduce subsequent PAC currents activated by a lower pH (*Figure 1—figure supplement 1C, D*). This suggests that unlike ASICs, PAC does not exhibit steady-state desensitization (*Wu et al., 2019*). These key distinctions indicate that the PAC channel must have a different desensitization mechanism from ASICs.

## Association of H98 with E107/D109 in the ECD–TMD interface promotes PAC channel desensitization

At pH 4.0 when the PAC channel undergoes desensitization, we previously observed a non-conducting human PAC structure with large conformational changes in the TMD and the ECD–TMD

interface compared to the resting state (*Ruan et al., 2020*). Specifically, TM1 undergoes a dramatic rotational movement, changing its interaction partner from the TM2 of the same subunit to that of the adjacent one. The rearrangement of the ECD–TMD interface is highlighted by the movement of histidine at position 98, which is decoupled from the resting position and brought in proximity to several negatively charged residues, including E107, D109, and E250, which constitute the 'acidic pocket' (*Figure 2A*). More recently, Wang et al. reported several human PAC structures, including a closed structure at pH 4.5 very similar to ours (*Wang et al., 2022*). The authors claimed that H98 is too far to make direct contacts with any of the acidic amino acids at low pH. We thus carefully compared our low pH non-conducting PAC structure (PDBID 7JNC) with theirs (PDBID 7SQH), and found that the distances of H98 from E107 and D109 are 2.9 and 3.9 Å in their structure, respectively (*Figure 2A*). This indicates that H98 can readily establish stable salt-bridge interactions with E107 and D109 as we previously proposed (*Kumar and Nussinov, 2002*; *Ruan et al., 2020*). However, H98 is about 8 Å away from the side chain of E250 (*Figure 2A*), and is therefore unlikely to interact with the latter.

Because the PAC channel desensitizes below pH 4.6 (*Figure 1*), we hypothesize that the non-conducting PAC structures observed at both pH 4.0 and pH 4.5 constitute a desensitized state. To test this hypothesis, we investigated if H98 and the associated acidic residues (E107 and D109) are involved in PAC desensitization, as their interactions are characteristics of the non-conducting PAC structure at low pH. We quantitatively compared the desensitization of charge-reversing mutants of these residues with that of wild-type PAC at pH 4.0, which exhibited obvious desensitization. Interestingly, all three mutants (H98R, E107R, and D109R) showed a similar impairment of desensitization at pH 4.0 (*Figure 2B and C*), suggesting that H98 and E107/D109 in the ECD–TMD interface facilitate PAC desensitization.

The reduced desensitization of the charge-reversing mutants (E107R and D109R) was expected, because a positive charge may interfere their association with protonated H98 at low pH, thereby destabilizing the proposed desensitized conformation. To mimic the protonation of E107 and D109, we made single and double charge-neutralizing mutants by replacing these two residues with gluta-mine and asparagine, respectively. These mutants also exhibited decreased desensitization although to a lesser degree compared to the charge-reversing mutants, suggesting that protonation of E107 and D109 partially underlies the pH dependence of PAC desensitization (*Figure 2B and C*). The H98Q mutant also exhibited less channel desensitization (*Figure 2B and C*), indicating that the lack of positive charge in residue 98 may impair its association with E107 and D109. However, the reduced desensitization of H98R was not anticipated, given the arginine residue maintains a posi-tively charged side chain similar to a protonated histidine residue. To understand the mechanism by which the H98R mutant reduces channel desensitization, all-atom unbiased MD simulations were performed using PAC structures in the low-pH non-conducting state. We accrued a total of 1 µs simulation data for both the wild-type and H98R mutant. Because the conformation of H98 is char-acteristic of the desensitized state, we focused on the local dynamics of this key residue to study the relationship between its conformation and channel desensitization. Clustering analysis of the side chains revealed that R98 in the H98R mutant adopted a much broader spectrum of side-chain conformations compared to the wild-type H98 (*Figure 2D and E*), indicating an increased flexibility when H98 is replaced with an arginine. Accordingly, the minimum distance between R98 and E107 or D109 over the MD trajectory increased (*Figure 2F and G*), suggesting a reduced tendency for R98 to form stable interactions with E107 or D109 compared to H98. These results also suggest that, in addition to the positive charge, the side-chain flexibility affects the association of residue 98 to the neighboring acidic residues (E107/D109). Further supporting this, the pairwise non-bonded interac-tion energy is greatly reduced for R98–D109 when compared to H98–D109 (*Figure 2I*). A noticeable but lesser effect is also seen for R98–E107 and H98–E107 (*Figure 2H*). Together, our data suggest that the reduced desensitization phenotype of H98R may attribute primarily to the interaction of H/R98 with D109, whereas the interaction with E107 played only a secondary role. As both H98Q/R contribute to PAC channel desensitization, our results indicate that the exact biochemical properties of histidine at position 98, including the charge and side-chain flexibility, are essential to fulfill the function. The involvement of H98 and its associated acidic residues in PAC desensitization provides experimental evidence that the previously observed PAC closed structure at low pH most likely represents a desensitized state.

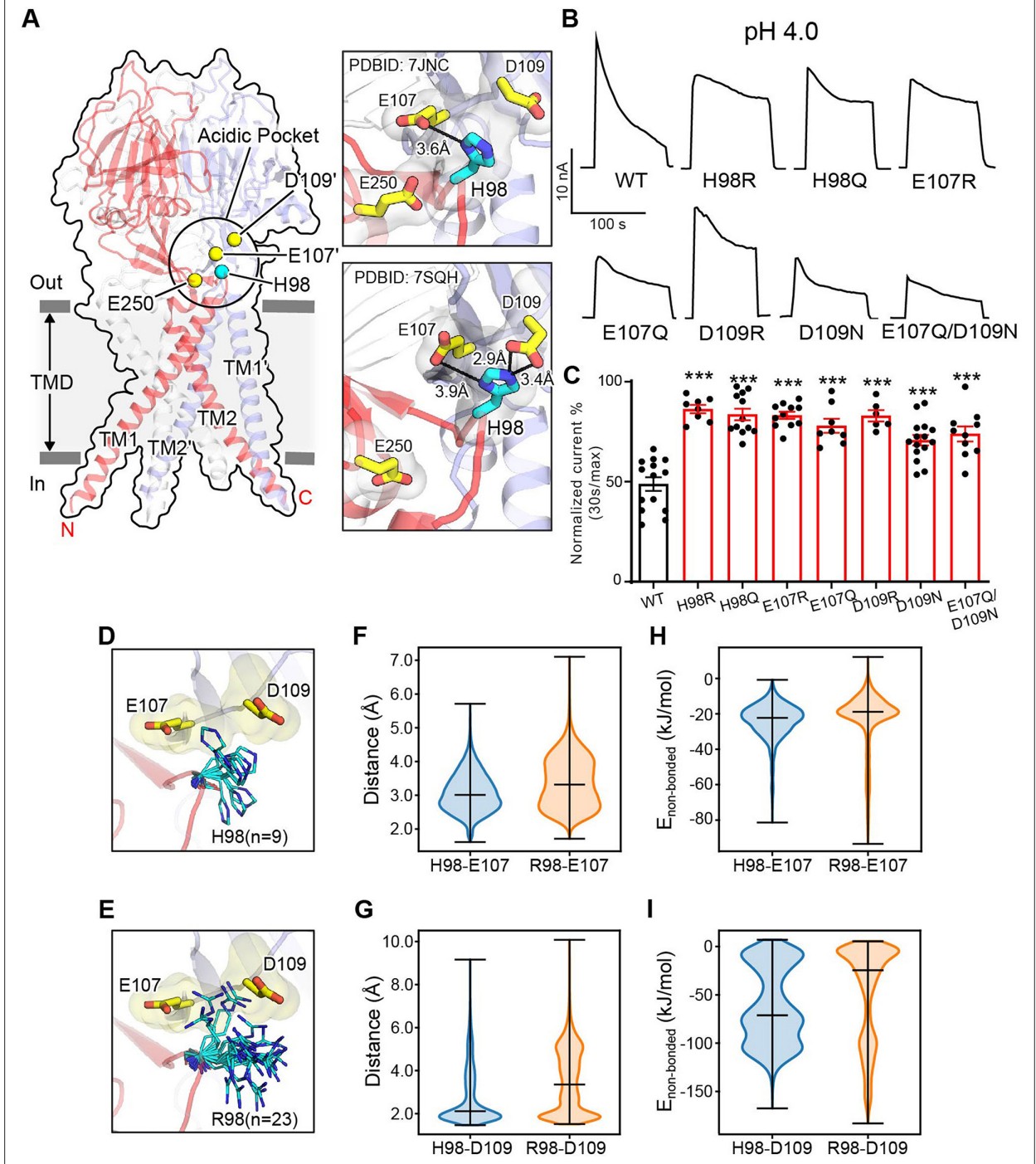

**Figure 2.** Association of H98 with E107/D109 in the extracellular domain–transmembrane domain (ECD–TMD) interface facilitates proton-activated chloride (PAC) channel desensitization. (**A**) The structure of human PAC at pH 4. The acidic pocket residues are shown in yellow spheres. Histidine 98 is shown in a cyan sphere. Close-up views of the local interactions at the acidic pocket are shown on the right. Two recently determined low-pH non-conducting human PAC structures (PDBID 7JNC and 7SQH) are compared. (**B**) Representative whole-cell current traces of wild-type PAC, and mutants (H98Q, H98R, E107R, E107Q, D109R, D109N, and E107Q/D109N) induced by extracellular pH 4.0 solution at +100 mV. (**C**) Percentage desensitization of wild-type PAC and mutants. Desensitized current after 30 s of pH 4.0 acidic solution exposure was normalized to the initial maximum current of that recording and expressed as a percentage. Data are mean ± SEM of the pH 4.0-induced currents at +100 mV. \*\*\*p<0.001; one-way analysis of variance (ANOVA) with Bonferroni post hoc test. (**D–E**) The local flexibility of H98 (D) and R98 (E) during molecular dynamics (MD) simulation. Clustering analysis is conducted for the side-chain atoms of H98/R98. The centroid conformation of each identified cluster is shown in the panels. A total of 9 clusters are identified for H98, whereas 23 clusters are found for R98, suggesting increased local flexibility of the H98R mutant. (**F**) The quantification of the minimum

*Figure 2 continued on next page*

*Figure 2 continued*

distance between the side-chain atoms of H98/R98 and E107 during the simulation. R98 samples a broader range of conformations relative to H98. (**G**) The quantification of the minimum distance between the side-chain atoms of H98/R98 and D109 during the simulation. R98 samples a broader range of conformations relative to H98. (**H**) The non-bonded interaction energy (Coulombic and Leonard-Jones potential) between H98/R98 with E107. The non-bonded energy associated with H98/E107 interaction is stronger when compared to R98/E107. (**I**) The non-bonded interaction energy (Coulombic and Leonard–Jones potential) between H98/R98 with D109. The non-bonded energy associated with H98/D109 interaction is stronger when compared to R98/D109. For panels (F-I), bars indicate median and extrema.

The online version of this article includes the following source data for figure 2:

**Source data 1.** Raw data for *Figure 2B-C, F-I*.

## E94 and D91 in the extracellular TM1 extension reduce PAC channel desensitization

During a comprehensive screen on the conserved acidic residues for potential pH sensors of PAC channel activation (*Osei-Owusu et al., 2022a*), we serendipitously discovered two additional residues (E94 and D91) that appear to play an opposite role in PAC desensitization compared to H98 and E107/D109 (*Figure 3A*). Replacing E94 with an arginine led to obvious desensitization at pH 5.0 (*Figure 3B and C*), when the wild-type PAC showed no current decay. Furthermore, at pH 4.6, the E94R mutant exhibited a greater degree of desensitization compared to the wild-type channel (*Figure 3B and D*). A positive charge at position 94 appears to be important for this phenotype as both arginine and lysine substitutions produced the effects, while replacing E94 with glutamine and aspartic acid had little to no influence on channel desensitization (*Figure 3B–D*).

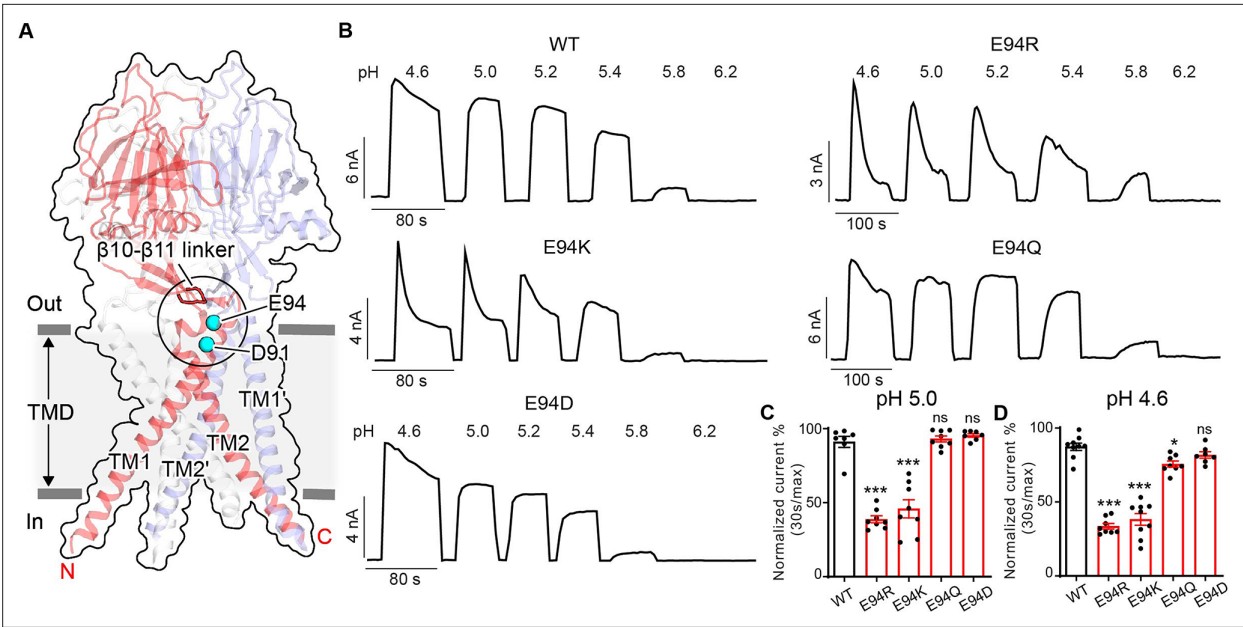

**Figure 3.** E94 in the extracellular transmembrane helix 1 (TM1) extension is critical for proton-activated chloride (PAC) channel desensitization. (**A**) The structure of human PAC at pH 4. The β10–β11 loop is shown in red. Glutamic acid (**E**) 94 and aspartic acid (**D**) 91 is shown in a cyan sphere. (**B**) Representative whole-cell current traces of wild-type PAC, and mutants E94R, E94K, E94Q, and E94D induced by different extracellular acidic pH solutions (4.6, 5.0, 5.2, 5.4, 5.8, and 6.2) at +100 mV. Cells were maintained in a pH 7.3 solution before applying acidic solutions. (**C**) Percentage desensitization of wild-type PAC, and mutants (E94R, E94K, E94Q, E94D) at pH 5.0. Desensitized current after 30 s of pH 5.0 acidic solution exposure was normalized to the initial maximum current of that recording and expressed as a percentage. Data are mean ± SEM of the pH 5.0-induced currents at +100 mV. ***$p<0.001$, ns, not significant; one-way analysis of variance (ANOVA) with Bonferroni post hoc test. (**D**) Percentage desensitization of wild-type PAC, and mutants (E94R, E94K, E94Q, E94D) at pH 4.6. Desensitized current after 30 s of pH 4.6 acidic solution exposure was normalized to the initial maximum current of that recording and expressed as a percentage. Data are mean ± SEM of the pH 4.6-induced currents at +100 mV. *$p<0.05$, ***$p<0.001$, ns, not significant; one-way ANOVA with Bonferroni post hoc test.

The online version of this article includes the following source data for figure 3:

**Source data 1.** Raw data for *Figure 3B–D*.

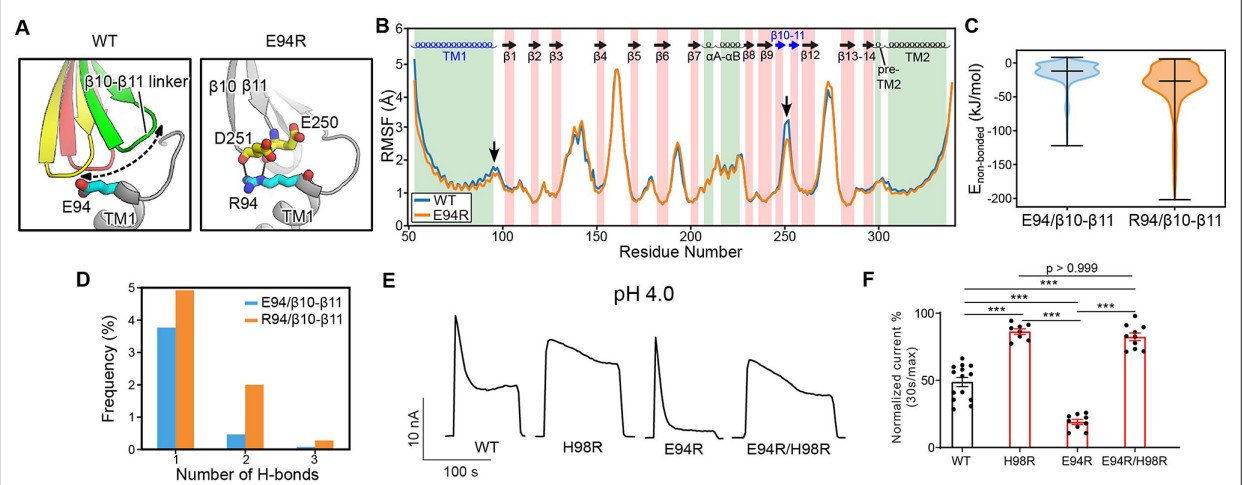

**Figure 4.** The E94R mutant promotes proton-activated chloride (PAC) channel desensitization by enhancing its interaction with the β10–β11 linker. (**A**) A schematic figure shows the impact of E94/R94 on the β10–β11 linker of human PAC at pH 4. (**B**) The Cα atom root-mean-square-fluctuation (RMSF) of wild-type (WT) and E94R mutant simulations. As indicated by black arrows, a decreased RMSF value is observed for the β10–β11 linker and TM1 for E94R mutant. (**C**) The non-bonded interaction energy (Coulombic and Leonard–Jones potential) between E94/R94 with the β10–β11 linker of PAC at pH4. Bars indicate median and extrema. (**D**) The number of hydrogen bonds between H94/R94 with β10–β11 loop. (**E**) Representative whole-cell current traces of WT PAC, and mutants (H98R, E94R, E94R/H98R) at pH 4.0 and +100 mV. Cells were maintained in pH 7.3 solution before applying acidic solutions. (**F**) Percentage desensitization of WT PAC, and mutants (H98R, E94R, E94R/H98R) at pH 4.0. Desensitized current after 30 s of pH 4.0 acidic solution exposure was normalized to the initial maximum current of that recording and expressed as a percentage. The WT datasets were pooled from several experiments and shared with *Figures 2C and 6C*. Data are mean ± SEM of the pH 4.0-induced currents at +100 mV. \*\*\*p<0.001; one-way analysis of variance (ANOVA) with Bonferroni post hoc test.

The online version of this article includes the following source data for figure 4:

**Source data 1.** Raw data for *Figure 4B–F*.

E94 is located at the C-terminus of the long TM1 helix, which extends into the extracellular space. When the channel adopts the desensitized conformation, E94 makes close contact with the β10–β11 linker at the ECD–TMD interface (*Figures 3A and 4A*). Therefore, we hypothesized that the interaction of E94 with the β10–β11 linker may affect the conformation of TM1 and thus the desensitized state of the channel. To test this idea, we performed MD simulations of the E94R mutant and compared the result with that of the wild-type protein. Consistent with our hypothesis, we noticed that the β10–β11 linker as well as TM1 and TM2 in the E94R mutant displayed attenuated dynamics compared to the wild-type as quantified by the root-mean-square-fluctuation (RMSF) analysis (*Figure 4B*), suggesting a decreased local flexibility caused by the E94R mutation. Accordingly, the non-bonded interaction energy between the residue 94 and the β10–β11 loop is strengthened in the E94R mutant (*Figure 4C*). We also analyzed the hydrogen bonding interaction between E94/R94 with the β10–β11 loop. Although the hydrogen bonds are only established in a subset of snapshots (~10%) during the simulations, R94 established more hydrogen bonds and interacted more frequently with the β10–β11 loop than E94 (*Figure 4D*). Therefore, the simulation results support the experimental observation that a positively charged residue at position 94, such as R94, could promote the fast-desensitizing phenotype (*Figure 3B–D*). This is likely because the favorable interaction of R94, at the extracellular TM1 extension, with the β10–β11 linker may stabilize TM1 in the desensitized conformation (*Figure 4A*).

We next wanted to understand the compound effect of E94 and H98, as both are located at the C-terminus of the long TM1 helix and affect PAC desensitization through their local interactions, but in opposite ways. Therefore, we generated the E94R/H98R double mutant and found that it reversed the fast-desensitizing phenotype of the single E94R mutant with diminished desensitization at pH 4.0 (*Figure 4E and F*), similar to that of the single H98R mutant. Taken together, these data suggest that the interactions of H98 with E107/D109 play a dominant role in promoting desensitization, while the E94R mutant is unable to facilitate desensitization when H98–E107/D109 interactions are weakened or disrupted.

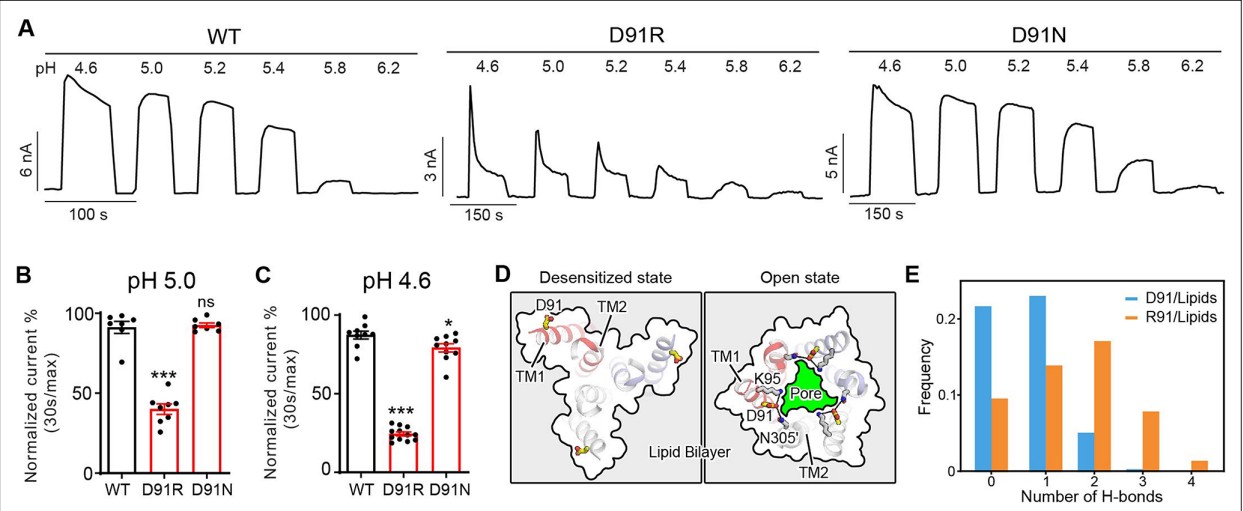

**Figure 5.** D91 reduces proton-activated chloride (PAC) channel desensitization by destabilizing transmembrane helix 1 (TM1) and lipid interaction. (**A**) Representative whole-cell current traces of wild-type (WT) PAC, and mutants D91R, and D91N induced by different extracellular acidic pH solutions (4.6, 5.0, 5.2, 5.4, 5.8, and 6.2) at +100 mV. The WT trace is the same as in *Figure 3B*. Cells were maintained in pH 7.3 solution before applying each acidic solution. (**B**) Percentage desensitization of WT PAC, and mutants D91R, and D91N at pH 5.0. Desensitized current after 30 s of pH 5.0 acidic solution exposure was normalized to the initial maximum current of that recording and expressed as a percentage. Data are mean ± SEM of the pH 5.0-induced currents at +100 mV. \*\*\*p<0.001, ns, not significant; one-way analysis of variance (ANOVA) with Bonferroni post hoc test. (**C**) Percentage desensitization of WT PAC, and mutants D91R, and D91N at pH 4.6. Desensitized current after 30 s of pH 4.6 acidic solution exposure was normalized to the initial maximum current of that recording and expressed as a percentage. The WT data were collected in the same set of experiments and shared with *Figure 3C and D*. Data are mean ± SEM of the pH 4.6-induced currents at +100 mV. \*p<0.05, \*\*\*p<0.001; one-way ANOVA with Bonferroni post hoc test. (**D**) Comparison of the D91 conformation between the desensitized and open states. D91 is facing the lipid bilayer in the desensitized state, whereas it is pointing towards the ion-conducting pore in the open state. Mutation of D91R may destabilize the polar interactions with K95 and N305, resulting in destabilization of the open state. (**E**) The number of hydrogen bonds between D91/R91 with lipid molecules. R91 makes more favorable hydrogen-bonding interactions with lipid head group, which contribute to the stabilization of the desensitized state.

The online version of this article includes the following source data for figure 5:

**Source data 1.** Raw data for *Figure 5A–C and E*.

Another interesting mutant we identified during the screen was an arginine substitution of D91, which is located approximately one helical turn below E94, on the extracellular face of TM1 helix (*Figure 3A*). Similar to E94R, the charge-reversing mutant of this residue, D91R, resulted in strong desensitization at pH 5.0 when the wild-type PAC channel exhibited no desensitization (*Figure 5A and B*). Furthermore, the D91R mutant showed a larger desensitization with a steeper kinetics compared to the wild-type at pH 4.6 (*Figure 5A and C*). However, this fast-desensitizing effect was not observed in the charge-neutralizing mutant, D91N (*Figure 5A–C*). These results suggest that like E94, a positive charge at position 91 also seems to be important for this phenotype. We were initially puzzled by how the D91R mutant contributes to channel desensitization as this residue appears to face the lipid bilayer in the low-pH desensitized structure and does not form any interaction with the rest of the protein. This is confirmed by MD simulations, which showed that D91 is indeed positioned toward the lipid bilayer in the desensitized conformation (*Figure 5D*). The simulation data further showed that R91 established more hydrogen bonds with the hydrophilic groups of the lipid bilayer and interacted with them more frequently relative to D91 (*Figure 5E*). The enhanced hydrogen bonding interactions with lipids may explain why the desensitized state becomes more prevalent for D91R. Moreover, in the recently determined open state structure of PAC (*Wang et al., 2022*), D91 points toward K95 and N305 near the extracellular opening of the ion conducting pore (*Figure 5D*). Of note, the cysteine substitution of N305 was found to dramatically reduce the PAC currents (*Ullrich et al., 2019*; *Yang et al., 2019*), highlighting the importance of the side-chain conformation of K95 and N305 in channel gating. Therefore, in addition to stabilizing PAC desensitized conformation, the D91R mutation may also alter its interactions with K95 and N305, thereby contributing negatively to the stability of the channel in the open state. Together, the fast-desensitizing phenotype observed

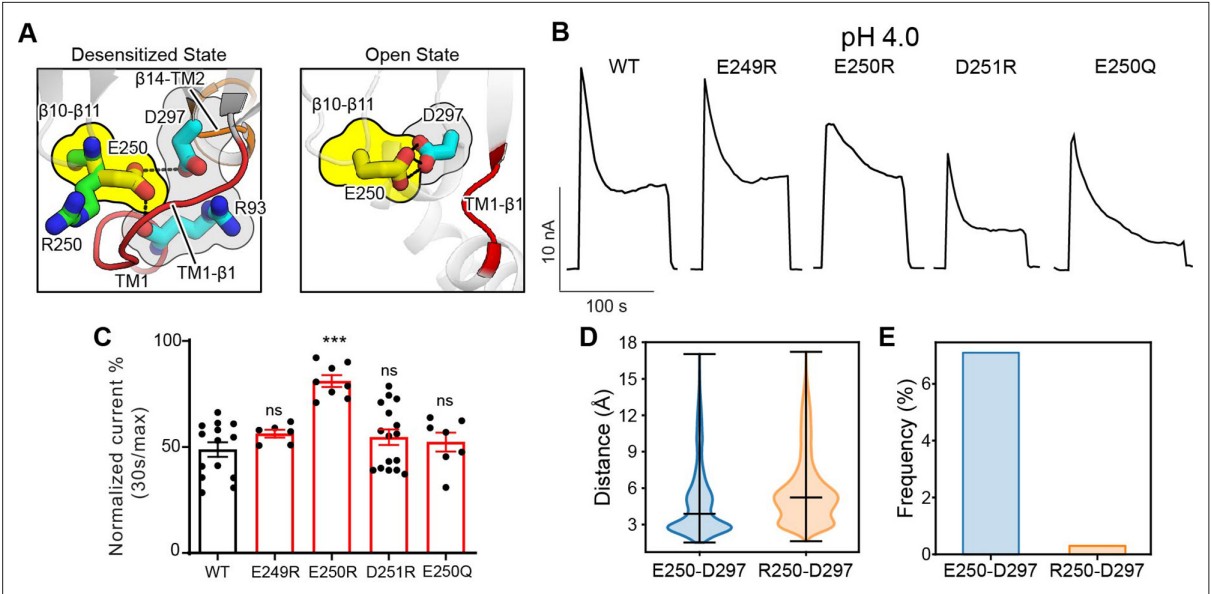

**Figure 6.** The β10–β11 linker regulates proton-activated chloride (PAC) channel desensitization via interactions with the transmembrane domain (TMD). (**A**) Interactions between E250 in β10–β11 linker with TM1–β1 linker (**R93**) and β14–TM2 linker (D297). The left panel shows the desensitized state of PAC in which an arginine substitution at position 250 would result in its side chain pointing away from the pocket formed by the TM1–β1 and β14–TM2 linkers. The right panel shows the recently determined open state structure of PAC in which TM1–β1 adopts a different conformation which may pose less restriction to the side-chain conformation of E250/R250. (**B**) Representative whole-cell current traces of wild-type PAC and mutants (E249R, E250R, D251R, E250Q) at pH 4.0 and +100 mV. Cells were maintained in pH 7.3 solution before applying acidic solutions. (**C**) Percentage desensitization of wild-type PAC and mutants (E249R, E250R, D251R, E250Q) at pH 4.0. Desensitized current after 30 s of pH 4.0 acidic solution exposure was normalized to the initial maximum current of that recording and expressed as a percentage. Data are mean ± SEM of the pH 4.0-induced currents at +100 mV. ***$p<0.001$, ns, not significant, one-way analysis of variance (ANOVA) with Bonferroni post hoc test. (**D**) The minimum distance between E250/R250 with D297 during molecular dynamics (MD) simulation. E250R showed a larger distance indicating a weaker interaction with D297. Bars indicate median and extrema. (**E**) The frequency of forming hydrogen bonds between E250/R250 with D297. Due to the side-chain occlusion, R250 seldom makes hydrogen bond with D297.

The online version of this article includes the following source data for figure 6:

**Source data 1.** Raw data for *Figure 6B–E*.

for the charge-reversing mutations of E94 and D91, both located in the extracellular TM1 extension, highlights the importance of this region in PAC channel desensitization.

## The β10–β11 linker is another regulator of PAC channel desensitization

The β10–β11 linker is part of the ECD–TMD interface and contains several negatively charged residues, including E249, E250, and D251. Among them, only E250 is involved in intra-subunit interactions in the desensitized structure (PDBID: 7SQH), where it is inserted into a pocket surrounded by the TM1–β1 and β14–TM2 linkers (*Figure 6A*). To test if the acidic residues on the β10–β11 linker affect PAC desensitization, we studied the charge-reversing mutants of these residues. Interestingly, E250R but not E249R and D251R exhibited reduced PAC desensitization at pH 4.0 (*Figure 6B and C*). However, glutamine substitution at position 250 had no effect, suggesting that protonation of E250 is not involved in the pH-dependent PAC desensitization. To understand why E250R mutant results in reduced PAC desensitization, we inspected the local interactions of E250 in the structure. Simple in silico mutagenesis revealed that, unlike a glutamic acid, an arginine residue at position 250, due to its bulkier side chain, could not fit into the pocket without steric clashes with the TM1–β1 and β14–TM2 linkers in the desensitized confirmation (*Figure 6A*). Indeed, MD simulations showed that R250 tends to point its side chain away from the pocket compared to E250, thus maintaining a longer distance and thus weaker interactions with D297 in the β14–TM2 linker (*Figure 6A and D*). Likewise, the hydrogen-bonding occupancy between R250 and D297 is almost diminished in the E250R mutant (*Figure 6E*). As a result, the association of the β10–β11 linker with the TMD may be weakened in the E250R mutant, thus making the desensitized conformation less favorable. Moreover, we noticed that

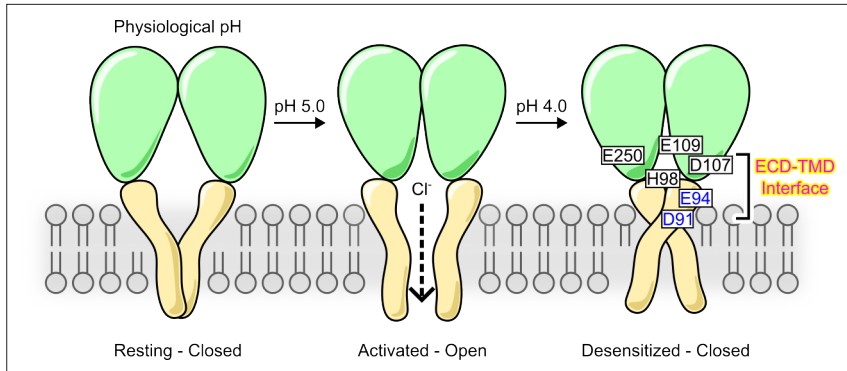

**Figure 7.** A schematic model showing proton-activated chloride (PAC) channel pH-dependent activation and desensitization. For simplicity, PAC trimer is depicted as two subunits, in the closed, open, and desensitized states. Excess extracellular acidification leads to PAC desensitization as a result of remodeling of the transmembrane domain (TMD). This process is regulated by critical interactions at the extracellular domain (ECD)–TMD interface, involving residues H98, E94, and D91 at the extracellular extension of the transmembrane helix 1 (TM1) and E107, D109, and E250 at the ECD–TMD interface. These interactions likely facilitate PAC desensitization by stabilizing the desensitized conformation of TM1. Therefore, our structure–function analysis supports the proposal that the previously observed proton-bound non-conducting structures of PAC represent a desensitized state of the channel (*Ruan et al., 2020*; *Wang et al., 2022*). Permeation of Cl⁻ ions in the activated open state is illustrated.

E250 also makes intimate interaction with D297 in the open state structure of PAC (PBDID 7SQF), which may be important for maintaining the open conformation of TM2 (*Figure 6A*). Therefore, in the E250R mutant, the salt bridge between R250 and D297 may enhance the interaction between these two residues, making the channel less susceptible to transition from the open to desensitized state. Taken together, E250 represents an important tethering point between β10–β11 linker and the TMD, which regulate PAC channel desensitization.

## Discussion

In this study, we provided a comprehensive characterization on the desensitization property of PAC, a newly identified ion channel activated by extracellular acid. By combining functional studies with MD simulations, we further revealed various molecular mechanisms underlying PAC desensitization (*Figure 7*). Previous cryo-EM studies reported a proton-bound non-conducting PAC structure at low pH (4.0 and 4.5), which was proposed as a desensitized state (*Ruan et al., 2020*; *Wang et al., 2022*). While the ECD of the suggested desensitized state adopts a similarly contracted conformation as that in the activated structure, the TMD (especially TM1) and the ECD–TMD interface undergo significant remodeling. Here, we showed that mutations in the extracellular extension of TM1 and their potential interacting partners in the ECD–TMD interface dramatically altered the degree and kinetics of PAC channel desensitization. Specifically, mutations predicted to destabilize the low-pH non-conducting structure (H98R, E107R, D109R, and E250R) greatly reduced channel desensitization; on the other hand, those predicted to stabilize it (E94R and D91R) produced the opposite effect (*Figure 7*). Thus, our structure-function analysis provided compelling evidence that the two similar proton-bound non-conducting structures previously observed at pH 4.0 and 4.5 indeed represents a desensitized state of the PAC channel.

During the transition from the activated to the desensitized conformation, TM1 of the PAC homo-trimer undergoes a dramatic rotational movement, switching its interaction partner from its cognate TM2 helix to the TM2 of a neighboring subunit (*Ruan et al., 2020*). This unusually large conformational change within the lipid membrane likely requires the PAC channel to overcome a high energy barrier to become desensitized. Consistently, the PAC currents do not show regular or steady-state desensitization under mildly acidic conditions. They start to desensitize only after reaching the maximal channel activity at pH 4.6. The desensitization kinetics is rather slow with a significant fraction of currents remaining even at pH 3.6. Furthermore, a brief exposure to neutral pH is sufficient to fully recover the PAC channel from the previous desensitization (*Figure 1—figure supplement 1A, B*).

Therefore, the characteristics of PAC desensitization are consistent with the large conformational change observed for the transition from the activated state to the desensitized one. Such a channel desensitization mechanism is in sharp contrast with the related ASIC channels in which the desensitized conformation is overall very similar to the open state except with a collapsed pore (*Baconguis et al., 2014*; *Gonzales et al., 2009*; *Jasti et al., 2007*). Accordingly, the desensitization rate of the ASIC currents is about two orders of magnitude faster than that of the PAC currents (*Rook et al., 2020*; *Wu et al., 2019*; *Zhang and Canessa, 2002*).

In addition to desensitization, the defining property of the PAC channel is its pH-dependent activation and gating. At room temperature, the channel begins to activate at pH ~6.0, with $pH_{50}$ and maximal activation of ~5.1 and ~4.6, respectively (at 37°C, $pH_{50}$ shifts ~0.5 units to less acidic pH). Interestingly, PAC channel desensitization is also dependent on pH, albeit in the more acidic range. A modest desensitization starts from pH ~4.6 and progressively becomes more pronounced when pH further drops to 3.6. This pH range is very close to the side-chain pKa values (~4.0) of acidic residues (glutamic acid and aspartic acid). Thus, protonation of increasing numbers of acidic residues (such as E107 and D109) likely contributes to the pH dependence of PAC desensitization. Among all the mutants we examined in this study, E94R and D91R (replacing the acidic residues with the positively charged arginine) stood out with strong desensitization in the higher pH range (above pH 5.0) that does not induce desensitizing for the wild-type PAC channel. This unique 'gain-of-function' phenotype suggests that these two acidic residues might also contribute to the pH dependence of PAC desensitization. However, substituting them with glutamine and asparagine, respectively, which closely mimic their protonated forms, failed to change in the desensitization profile of PAC. Thus, while playing a critical role in preventing PAC channel desensitization, E94 and D91 in the extracellular side of TM1 may not be sufficient to mediate the pH sensitivity of PAC desensitization, at least individually. Future mutagenesis study will further identify critical titratable residues that regulate the pH dependence of PAC desensitization. Similar to the pH-sensing mechanism for the PAC channel activation (*Osei-Owusu et al., 2022a*), it's most likely that multiple titratable amino acids in the ECD and ECD–TMD interface, presumably with a lower pKa, collectively tune the conformational equilibrium of PAC from the open state to the desensitized state, making PAC desensitization pH-dependent.

Our study employed unbiased MD simulations to investigate the structural impact of mutations that affect PAC desensitization, which provided detailed mechanistic insight to rationale our experimental observation. However, our strategy also bears caveats that are worth pointing out. First, due to the limitation on modeling bond formation and bond breaking events in conventional MD, we only investigated the predominant protonation form of the relevant residues. This is far from ideal as many relevant residues (His/Asp/Glu) may undergo protonation and deprotonation events depending on their side-chain orientation and the surrounding electrostatic environment, which might be critical for their function. As an example, our electrophysiology data suggest that D109N mutant results in slow-desensitizing phenotype, suggesting that a protonated D109 is probably involved in the pH-dependent desensitization process. However, our simulation only considered a de-protonated form of D109 due to its predicted pKa being below 4.0 (*Table 1*). Second, the accessible simulation timescale is also very limited, preventing us from capturing the intermediate conformations of PAC from the desensitized state to the open and resting state. Our experimental evidence suggests that all the three states are inter-convertible, but the pH-dependent free energy landscape and transition pathway of this process is not clear. A better description of this process will for sure offer novel insight into the function and regulation of PAC channel. Future studies that combine constant pH MD simulations coupled with enhanced sampling approaches may help address these questions (*Martins de Oliveira et al., 2022*).

Future study is also required to establish the physiological relevance of PAC desensitization. Severe local acidosis causes tissue damage in many diseases, including cerebral and cardiac ischemia, cancer, infection, and inflammation (*Gatenby and Gillies, 2008*; *Lardner, 2001*; *Yang*

**Table 1.** The predicted residue pKa value of relevant residues for the wild-type (WT) and mutants investigated.

|  | WT | H98R | E94R | D91R |
|---|---|---|---|---|
| pKa(H98) | 6.10 | 12.42 (R) | 6.12 | 6.12 |
| pKa(E107) | 5.28 | 5.53 | 5.29 | 5.27 |
| pKa(D109) | 3.46 | 3.68 | 3.46 | 3.46 |
| pKa(E250) | 4.76 | 4.94 | 4.78 | 4.78 |
| pKa(E94) | 4.84 | 4.79 | 12.53 (R) | 4.52 |
| pKa(D91) | 3.70 | 3.69 | 3.59 | 12.45 (R) |

*et al., 2019*). It's possible that channel desensitization protects the cells from injury and death during the long exposure of tissue acidosis by reducing PAC-mediated Cl⁻ influx and subsequent cell swelling (*Osei-Owusu et al., 2020*; *Wang et al., 2007*; *Yang et al., 2019*). Although this is consistent with the idea of receptor desensitization serving as a protective mechanism in the setting of aberrant signaling, overall tissue pH in various pathologies rarely drops to pH 6.0 or below. Alternatively, lysosomes, which can be acidified down to pH 4.5, may present a physiological environment for PAC desensitization. During endolysosomal acidification, Cl⁻, as the counterion, provides the electrical shunt for proton pumping by the vacuolar $H^+$ ATPase (*Mindell, 2012*; *Stauber and Jentsch, 2013*). PAC normally traffics to endosomes where it functions as a low pH sensor and prevents hyper-acidification by releasing Cl⁻, from the lumen, thus maintaining endosomal pH in an optimal range. PAC does not appear to co-localize with the lysosomal marker LAMP1 (*Osei-Owusu et al., 2021*). However, even if a fraction of the PAC channel traffics further to lysosomes, the low pH environment may induce PAC desensitization and prevent it from leaking Cl⁻ from the lysosomal lumen. This potentially provides an additional mechanism to maintain the higher inside Cl⁻ concentration and increased acidity in lysosomes compared to endosomes. Our work on the structural mechanism underlying PAC channel desensitization builds the molecular foundation for further exploring its role in cellular physiology and disease.

# Materials and methods

## Key resources table

| Reagent type (species) or resource | Designation | Source or reference | Identifiers | Additional information |
|---|---|---|---|---|
| cell line (*Homo-sapiens*) | HEK293T | ATCC | Cat#: CRL-3216 | |
| cell line PACC1 knockout (*Homo-sapiens*) | PAC KO HEK293T | doi:10.1126/science.aav9739 | | |
| recombinant DNA reagent | pIRES2-EGFP-hPAC | doi:10.1126/science.aav9739 | | |
| commercial assay or kit | Lipofectamine2000 | Invitrogen | Cat#: 11668019 | |
| commercial assay or kit | QuikChange II XL site-directed mutagenesis | Agilent Technologies | Cat#: 200522 | |
| software, algorithm | pCLAMP 10.7 | Molecular Devices | RRID:SCR_011323 | |
| software, algorithm | Clampfit 10.7 | Molecular Devices | | |
| software, algorithm | PyMOL 2.3.4 | Schrodinger, LLC | | |
| software, algorithm | Gromacs 2020.1 | doi: 10.1016/j.softx.2015.06.001 | | |
| software, algorithm | CHARMM-GUI | doi: 10.1002/jcc.23702 | | |
| software, algorithm | PPM Web Server | doi: 10.1093/nar/gkr703 | | |
| software, algorithm | Python | Python Software Foundation | | |
| software, algorithm | GraphPad Prism 8 | GraphPad | | |

## Cell culture

HEK293T cells were purchased from ATCC and routinely maintained in the laboratory. Although the cell lines were not further authenticated experimentally after purchase, we verified that these are the same cells based on growth and morphology. They tested negative for mycoplasma contamination and were maintained in Dulbecco's modified Eagle's medium (DMEM) supplemented with 10% fetal bovine serum (FBS) and 1% penicillin/streptomycin (P/S) at 37°C in humidified 95% $CO_2$ incubator. *PAC* KO cells were previously generated using CRISPR technology (*Yang et al., 2019*), and maintained

in DMEM supplemented with 10% FBS and 1% P/S at 37°C in humidified 95% $CO_2$ incubator. *PAC* KO cells were transfected with 800–1000 ng/ml of plasmid DNA using Lipofectamine 2000 (Life Technologies) according to the manufacturer's instruction. Cells were seeded on 12 mm diameter poly-D-lysine coated glass coverslips (BD) and were recorded ~1 day later after transfection.

## Constructs and mutagenesis

The coding sequence of human PAC (NP_060722) previously subcloned into pIRES2-EGFP vector (Clontech) using XhoI and EcoRI restriction enzyme sites (*Yang et al., 2019*) was used for whole-cell patch-clamp recording. Site-directed mutagenesis were introduced into plasmids using QuikChange II XL site-directed mutagenesis kit (Agilent Technologies) and confirmed by Sanger sequencing.

## Electrophysiology

Whole-cell patch-clamp recordings were performed as previously described (*Osei-Owusu et al., 2022a*; *Yang et al., 2019*). Wild-type HEK293T and PAC KO cells were plated on poly-D-lysine-coated coverslips for ~1 day before recording. The KO cells were transfected with plasmids of human PAC or mutants. Cells were recorded in an extracellular solution (ECS) containing (in mM): 145 NaCl, 1.5 $CaCl_2$, 2 $MgCl_2$, 2 KCl, 10 HEPES, and 10 glucose (pH adjusted to pH 7.3 with NaOH and osmolality was 300–310 mOsm/kg). Acidic ECS was made with the same ionic composition without HEPES but with 5 mM $Na_3$-citrate as buffer and the pH was adjusted using citric acid. Solution exchange was achieved by using a perfusion system with a small tip 100–200 μm away from the recording cell. Recording pipettes with resistance 2–4 MΩ were filled with internal solution containing (in mM): 135 CsCl, 1 $MgCl_2$, 2 $CaCl_2$, 5 EGTA, 4 MgATP, and 10 HEPES (pH adjusted to 7.2 with CsOH and osmolality was 280–290 mOsm/kg).

All recordings were done at room temperature with MultiClamp 700B amplifier and 1550B digitizer (Molecular Devices). Data acquisition were performed with pClamp 10.7 software (Molecular Device), filtered at 2 kHz and digitized at 10 kHz. Proton-activated $Cl^-$ currents were recorded by applying voltage-ramp pulses every 3 s from –100 to +100 mV at a speed of 1 mV/ms, and at a holding potential of 0 mV. For desensitization quantification, pH-induced (5.0, 4.6, 4.0, or 3.6) currents measured at +100 mV were normalized by taking a ratio of the desensitized current amplitude at 30 s ($I_{30s}$) after acidic exposure to the peak amplitude ($I_{max}$) of that recording. This was then expressed as a percentage: $100 \times (I_{30s}/I_{max})$. To measure desensitization time constant, desensitized currents were normalized to the initial maximum current and analyzed by fitting to GraphPad prism one phase decay equation ($Y=(Y_0 - \text{Plateau})*\exp(-K*X)+\text{Plateau}$) to estimate the time constant (tau), where $Y_0$ is the Y value when X (time) is zero, Plateau is the Y value at infinite times, and K is the rate constant. Steady-state desensitization was measured by perfusing cells with a control (pH 7.3) and test (pH 5.8) conditioning solutions for 30 s before applying the pH 4.0 acidic solution. The initial maximum amplitudes of the first (1) and second (2) pH 4.0-induced currents after conditioning were compared to assess the fraction of channels that desensitize after the test conditioning. Recovery from desensitization was recorded by perfusing cells with a pH 4.0 acidic solution for at least 60 s, followed by a recovery pH at 7.3 for 5–10 s, and then a second application of pH 4.0 solution. The peak current amplitudes of the first (1) and second (2) pH 4.0-induced currents were then compared to assess the fraction of channels that had desensitized to the closed state after applying the pH 7.3 solution. All data were analyzed using Clampfit 10.7 and GraphPad Prism 8 software was used for all statistical analyses.

## MD simulations

PAC structure in desensitized state (PDBID: 7JNC) is used as the starting conformation for the simulation (*Ruan et al., 2020*). The protein is oriented with respect to membrane normal using the OPM webserver (*Lomize et al., 2012*). CHARMM-GUI is used to generate the membrane bilayer system for simulation (*Jo et al., 2008*). During the system setup, the protonation status of titratable residues is determined based on propka 3.0 (*Table 1*) and is assigned to the predominant protonation state at pH 4.0 (*Olsson et al., 2011*). Point mutations including H98R, E94R, and D91R are generated during the system preparation in CHARMM-GUI. The residue pKa of the mutants are re-evaluated using propka 3.0 (*Table 1*), and the protonation status of titratable residues are then assigned to their predominant state at pH 4.0. The membrane bilayer consists of POPC/POPE/cholesterol in 1:1:2 ratio. Protein is dissolved in TIP3P water model and the system charge is neutralized with approximated

150 mM NaCl. The simulation system contains about 157 k atoms with box size of 92Å × 92Å × 180 Å. CHARMM36m force field is used to parameterize the system (*Huang et al., 2017*). LINCS algorithm is used to constrain hydrogen atoms (*Hess, 2008*). The Verlet scheme is used to determine the neighbor list for calculating short-range non-bonded interactions with a cut-off of 12 Å. GROMCS v2020.1 is used as the simulation engine (*Abraham et al., 2015*). Energy minimization is conducted using steepest descent method such that the Fmax is less than 1000 kJ/mol. We conducted three rounds of equilibration in canonical ensemble by bringing the system temperature to 310 K using Berendsen thermostat (*Berendsen et al., 1984*). The protein, membrane, and solute atoms are assigned to separate atom groups for temperature coupling. A timestep of 1 ps is used for equilibration runs at canonical ensemble with heavy atom position restraint gradually relaxed. Subsequently, Berendsen barostat is enabled to keep the system pressure to 1 bar (*Berendsen et al., 1984*). Position restraint is further relaxed during the isothermal–isobaric ensemble runs with an integration step of 2 ps. Production simulation is conducted in isothermal–isobaric ensemble without position restraint. Five independent simulation runs are carried out for each condition (WT, H98R, E94R, D91R, and E250R) for 200 ns. As a result, the total sampling time is 1 µs for WT and each mutant. We combined the MD trajectories for data analysis. Because PAC is a trimeric complex, we combined the statistics obtained for each subunit of PAC and reports the summarized result. Analyses of the MD data, including the residue clustering, distance measurement, hydrogen bond detection, and energy calculation, are conducted using the build-in routines of GROMACS (*Abraham et al., 2015*). Results are visualized using custom Python script powered by matplotlib and seaborn library (code is available in GitHub; *Osei-Owusu et al., 2022b*; copy archived at swh:1:rev:8587d188301e271f01f03c7b0f259dd505de0abb). PAC structures are visualized using PyMOL v2.5 (*Schrödinger and DeLano, 2020*).

## Quantification and statistical analysis

Data and statistical analyses were performed using Clampfit 10.7, GraphPad Prism 8, and Excel. Statistical analyses between two groups were performed using two-tailed Student's *t* test, unless reported otherwise. Multiple-group comparisons were performed using one-way analysis of variance (ANOVA) with Bonferroni post hoc test. Non-significant 'ns' p values were not reported. All numeric data are shown as mean ± SEM and numbers per group are represented in bars or legends of each panel. The significance level was set at $p<0.05$.

## Acknowledgements

JO-O is supported by an American Heart Association (AHA) predoctoral fellowship (grant 18PRE34060025). ZR is supported by an AHA postdoctoral fellowship (grant 20POST35120556) and the National Institute of Health (grant K99NS128258). LM is supported by Boehringer Ingelheim Fonds (BIF) and National Institute of General Medical Sciences, T32 GM007445 (to the BCMB graduate training program). DSM is supported by a Physician Scientist Training Program grant from Johns Hopkins University School of Medicine. WL is supported by the National Institute of Health (NIH) (grant R01NS112363). ZQ is supported by a McKnight Scholar Award, a Klingenstein-Simon Scholar Award, a Sloan Research Fellowship in Neuroscience, and NIH (grant R35GM124824 and R01NS118014).

## Additional information

### Funding

| Funder | Grant reference number | Author |
| --- | --- | --- |
| National Institutes of Health | R35GM124824 | Zhaozhu Qiu |
| National Institutes of Health | R01NS118014 | Zhaozhu Qiu |
| National Institutes of Health | R01NS112363 | Wei Lü |

| Funder | Grant reference number | Author |
|---|---|---|
| American Heart Association | 18PRE34060025 | James Osei-Owusu |
| American Heart Association | 20POST35120556 | Zheng Ruan |
| National Institutes of Health | K99NS128258 | Zheng Ruan |
| National Institute of General Medical Sciences | GM007445 | Kevin Hong Chen |

The funders had no role in study design, data collection and interpretation, or the decision to submit the work for publication.

### Author contributions

James Osei-Owusu, Zheng Ruan, Conceptualization, Data curation, Formal analysis, Writing – original draft, Writing – review and editing, Visualization; Ljubica Mihaljević, Data curation, Formal analysis, Writing – review and editing; Daniel S Matasic, Writing – original draft, Writing – review and editing; Kevin Hong Chen, Data curation, Writing – review and editing, Formal analysis; Wei Lü, Zhaozhu Qiu, Conceptualization, Formal analysis, Supervision, Funding acquisition, Writing – original draft, Writing – review and editing

### Author ORCIDs

James Osei-Owusu http://orcid.org/0000-0002-8676-4444
Zheng Ruan http://orcid.org/0000-0002-4412-4916
Wei Lü http://orcid.org/0000-0002-3009-1025
Zhaozhu Qiu http://orcid.org/0000-0002-9122-6077

### Decision letter and Author response

Decision letter https://doi.org/10.7554/eLife.82955.sa1
Author response https://doi.org/10.7554/eLife.82955.sa2

## Additional files

### Supplementary files

• Transparent reporting form

### Data availability

All data generated or analysed during this study are included in the manuscript and supporting files.

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
