## [Editor Report]

This important study addresses the molecular mechanisms of the proton-activated chloride channel (PAC), a widely expressed ion channel involved in organelle pH homeostasis and acid-induced cell death. Convincing data based on structure-guided mutagenesis and molecular dynamics simulations provides new insight into the mechanism underlying channel desensitization under sustained acidic stimulation. The results are of interest to ion channel physiologists.

---

## [Decision Letter]

**Decision letter after peer review:**

Thank you for submitting your article "Molecular mechanism underlying desensitization of the proton-activated chloride channel PAC" for consideration by *eLife*. Your article has been reviewed by 3 peer reviewers, including László Csanády as Reviewing Editor and Reviewer #1, and the evaluation has been overseen by Richard Aldrich as the Senior Editor.

Essential revisions:

1. The model would be much more complete if specific protonation events involved were identified or at least clearly hypothesized. Can one say definitively that additional protonation events after opening are required as indicated in the model figure? Are they on residues implicated here? In addition to the charge reversal mutations presented, charge neutralizing mutations could be evaluated to address this: (1) E107Q, (2) D109N, and (3) E107Q/D109N to explore protonation of the acidic pocket and (4) E249Q, (5) E250Q, and (6) D251N to explore protonation of the beta10-11 linker.

Related to the above, the model figure 7 would benefit greatly if protonation states of residues (or groups of residues) were indicated, especially as previous work has implicated residues in different processes. In addition, predicted pKas in wild-type and mutant structures for the relevant acidic residues could be presented to guide possible protonation mechanisms and explain mutant effects.

2. The MD simulations raise several technical concerns, and are often inconsistent with the functional results. These issues should be addressed or at least acknowledged:

2.1. The MD analysis critically depends on assumptions on the protonation states of multiple residues, that are often located in close proximity to each other. In the methods, the authors state they use PropKa to estimate the pKa of residues and assigned the protonation states based on this. What pH was considered in the simulations? Was the propKa analysis run considering how choices in the protonation state of neighboring residues affect the pKa of the other residues? This is critical because the interaction energies will greatly depend on the protonation state chosen. Was the pKa for the mutant constructs re-evaluated? For example, does having a Gln or Arg in place of a His affect the pKa of nearby acidic residues? (Altered desensitization in H98R could be explained by perturbed pKa of nearby E107/D109 in addition to (or instead of) reduced interaction strength between the 98-109 side chains.)

2.2. The experimental data suggests that H98, E107, and D109 play analogous roles in PAC desensitization. However, the MD simulations suggest that the H98-D109 interaction energy is ~4 times larger than that of H98-E107. This should lead to a much greater effect of the D109 mutation. How is this rationalized?

2.3. The experimental data shows that E94 plays a key role in desensitization and the authors argue that this is due to the interactions of this residue with the β10-11 linker. However, the MD simulations show that these interactions happen for a small fraction, ~10%, of the time and with interaction energies comparable to those of the H98-E107-D109 cluster. It is not clear how these sparse and transient interactions can play such a critical role in desensitization. Also, if the interaction energies are of the same sign, how come one set of mutants favors desensitization and one does not?

2.4. Are 600 ns sufficient to evaluate sampling of the different conformations? The MD simulations should be extended beyond the 600 ns to increase sampling, ideally, multiple independent repeats should be carried out and analyzed at different pHs.

3. The underlying assumption in the interpretation of all the data is that the mutations stabilize or destabilize the desensitized conformation of the channel. However, none of the functional measurements provide direct evidence supporting this key assumption. The conclusions would be greatly strengthened if the authors could directly show that the mutations speed up or slow down the rate of recovery from the desensitized state. This should be feasible for most constructs as activation and desensitization happen at different pHs.

4. The reliability of the reported time constants of inactivation is questionable. For the reliable fitting of an exponential function, a time course at least 2-3x longer than the time constant itself must be fitted. In many cases, the presented time courses are too short to afford an exponential fit. E.g.: Figure 2B (H98R, E107R); Figure 3B (WT at pH 4.6 or 5.0, E94Q at pH 4.6 or 5.0); Figure 4E (H98R, E94R/H98R); Figure 5 (WT at pH 4.6 or 5.0, D91N at pH 4.6 or 5.0); Figure 6B (E250R).

*Reviewer #1 (Recommendations for the authors):*

1. I am concerned about the reliability of the reported time constants of inactivation. For the reliable fitting of an exponential function, a time course at least 2-3x longer than the time constant itself must be fitted. In many cases, the presented time courses are too short to afford an exponential fit. E.g.: Figure 2B (H98R, E107R); Figure 3B (WT at pH 4.6 or 5.0, E94Q at pH 4.6 or 5.0); Figure 4E (H98R, E94R/H98R); Figure 5 (WT at pH 4.6 or 5.0, D91N at pH 4.6 or 5.0); Figure 6B (E250R). The normalized surviving current fraction after 30 s is a simpler and more reliable parameter – maybe it would suffice for supporting the authors' conclusions, even if reliable time constants cannot always be extracted from the data.

*Reviewer #2 (Recommendations for the authors):*

Overall, this is a nice study with well-done experiments. Interpretations of the data seem largely reasonable, though some questions remain about exactly how to reconcile structural and functional data and the precise molecular mechanisms underlying desensitization. I have several comments that could improve the interpretability and accessibility of the study for a general audience.

1. The major issue is that important regions for desensitization are implicated when the model would be much more complete if specific protonation events involved were identified or at least clearly hypothesized. Can one say definitively that additional protonation events after opening are required as indicated in the model figure? Are they on residues implicated here? Several additional mutants could be evaluated to address this: (1) E107Q, (2) D109N, and (3) E107Q/D109N to explore protonation of the acidic pocket and (4) E249Q, (5) E250Q, and (6) D251N to explore protonation of the beta10-11 linker (in addition to charge reversal mutations presented). In addition, predicted pKas in wild-type and mutant structures for the relevant acidic residues could be presented to guide possible protonation mechanisms and explain mutant effects. For example, altered desensitization in H98R could be explained by perturbed pKa of nearby E107/D109 in addition to (or instead of) reduced interaction strength between the 98-109 side chains.

2. Related to 1, the model figure 7 would benefit greatly if protonation states of residues (or groups of residues) are indicated, especially as previous work has implicated residues in different processes.

3. I do not understand the explanation provided for the difference between the properties of endogenous PAC and overexpressed PAC in a PAC -/- cell line, which is substantial. Why should overexpression (resulting in ~4x current density) results in a higher degree and faster rate of desensitization? This is an interesting difference to point out even if the reason for the difference is unclear.

4. A brief discussion further connecting the structure papers with this work would help with context. How do the proportions of open, closed, and desensitized structural states at high and low pH values (refs 17,18, Deng et al. Sci. Adv. 2021) match the expected distribution based on the recordings here? Can the authors explain discrepancies in different conditions (e.g. detergent vs nanodisc)?

5. Line 212 appears to include a typo. Residue107-249 intersubunit and 250-297 intrasubunit interactions are observed in PDB 7SQH.

6. This sentence may be misworded. "Specifically, mutations predicted to stabilize the low-pH non-conducting structure (H98R, E107R, D109R, and E250R) greatly reduced channel desensitization; on the other hand, those predicted to destabilize it (E94R and D91R) produced the opposite effect (Figure 7)." H98R, E107R, D109R, and E250R should destabilize the desensitized state to reduce desensitization.

7. Figure 7 legend: does the model depict one or two subunits?

*Reviewer #3 (Recommendations for the authors):*

Suggestions:

I think it is critical to directly show that the mutations speed up or slow down the recovery from desensitization. This should be feasible for most constructs as activation and desensitization happen at different pHs.

The MD simulations should be extended beyond the 600 ns to increase sampling, ideally, multiple independent repeats should be carried out and analyzed at different pHs.

The effects of protonation choices on different should be evaluated.

The inconsistencies between electrophysiological results and MD simulations should be addressed and, if it is not possible to resolve them, then they should be acknowledged.

- I do not see a reduction of interaction energy and of distance with E107 in H98 vs 98R (Figure 2), as claimed in the text (pg. 7).

---

## [Author Response]

Essential revisions:1. The model would be much more complete if specific protonation events involved were identified or at least clearly hypothesized. Can one say definitively that additional protonation events after opening are required as indicated in the model figure? Are they on residues implicated here? In addition to the charge reversal mutations presented, charge neutralizing mutations could be evaluated to address this: (1) E107Q, (2) D109N, and (3) E107Q/D109N to explore protonation of the acidic pocket and (4) E249Q, (5) E250Q, and (6) D251N to explore protonation of the beta10-11 linker.Related to the above, the model figure 7 would benefit greatly if protonation states of residues (or groups of residues) were indicated, especially as previous work has implicated residues in different processes. In addition, predicted pKas in wild-type and mutant structures for the relevant acidic residues could be presented to guide possible protonation mechanisms and explain mutant effects.

We thank the reviewer for the suggestion. We generated the recommended mutants and performed new experiments (except E249Q and D251N for which the charge-reversal mutations had normal desensitization). Similar to E107R and D109R, the E107Q and D109N mutants exhibited reduced desensitization at pH 4.0 (Figure 2B, C). These results suggest that the protonation of these two residues may partially mediate the pH-dependence of PAC desensitization by weakening the association of H98 with E107/D109 in the ECD-TMD interface. The double mutant did not further decrease the current decay, consistent with the idea that they likely work through the same structural mechanism. In contrast, the E250Q mutant exhibited normal desensitization as WT PAC channel (Figure 6B, C). This indicates that the protonation of E250 does not underlie the pH-dependence of PAC desensitization.

Although our study clearly identified a set of residues that play a critical role in PAC channel desensitization, we believe these residues regulate this process through complex mechanisms. While the new data presented in this revision suggests that protonation of E107 and D109 can account for the pH-dependent PAC desensitization to some degree, it is unlikely the sole mechanism. Our Figure 7 intends to show that polar interactions encoded in the ECD-TMD interface are important for PAC desensitization, not just through the protonation events of these residues per se. Therefore, we have revised the cartoon in Figure 7 to make less emphasis on the protonation events. We hope that this will cause less confusion and convey a better summary of our study.

2. The MD simulations raise several technical concerns, and are often inconsistent with the functional results. These issues should be addressed or at least acknowledged:2.1. The MD analysis critically depends on assumptions on the protonation states of multiple residues, that are often located in close proximity to each other. In the methods, the authors state they use PropKa to estimate the pKa of residues and assigned the protonation states based on this. What pH was considered in the simulations? Was the propKa analysis run considering how choices in the protonation state of neighboring residues affect the pKa of the other residues? This is critical because the interaction energies will greatly depend on the protonation state chosen. Was the pKa for the mutant constructs re-evaluated? For example, does having a Gln or Arg in place of a His affect the pKa of nearby acidic residues? (Altered desensitization in H98R could be explained by perturbed pKa of nearby E107/D109 in addition to (or instead of) reduced interaction strength between the 98-109 side chains.)

In our initial manuscript, the pKa analysis was done based on the WT structure and the residue protonation status was assigned based on the predicted value assuming the environment pH is at 4.0. For example, if the residue side-chain pKa is above 4.0, then it will be protonated, otherwise, it will not. We agree with the reviewer that mutations on certain residues could alter the pKa of neighboring residues. To evaluate this impact, we carried out pKa prediction for all the mutant structures that we used for simulation (see Table S1). The pKa prediction for H98, E107, and D109 is summarized in Table 1.

As shown from the table, the H98R mutant only caused a small pKa upshift for both E107 and D109. At the pH range where desensitization is observed (pH 4.0-4.6), D109 will remain unprotonated, whereas E107 will be protonated. Therefore, we don’t think altered pKa of E107/D109 by H98R is a major contributor for its reduced desensitization phenotype.

2.2. The experimental data suggests that H98, E107, and D109 play analogous roles in PAC desensitization. However, the MD simulations suggest that the H98-D109 interaction energy is ~4 times larger than that of H98-E107. This should lead to a much greater effect of the D109 mutation. How is this rationalized?

The purpose of quantifying the interaction between H/R98 with E107 and D109 is to better dissect the mechanism by which H/R98 interacts with the acidic pocket residues. The result suggests that R98 has a reduced association with E107/D109 when compared to H98. It also suggests that D109 makes a more direct interaction with H/R98 when compared to E107. We acknowledge that this is not clear in our initial manuscript, and we have updated the text to better describe this result. However, this doesn’t imply that the desensitization phenotype of E107R should be less pronounced than D109R. Both E107R and D109R are expected to disrupt the integrity of the acidic pocket, thus resulting in diminished channel desensitization. It is worth pointing out that E107 played a more complex role as it was identified in our previous papers as one of the major proton sensors. The E107R mutant could allow the PAC channel to become more sensitive to acid-induced activation (Figure 4d-e in Ruan et al., *Nature*, 2020), further complicating its effect in desensitization. Taken together, we don’t think the E107/D109 and H/R98 interaction strength could have quantitative correlation with the desensitization phenotype of E107R and D109R.

2.3. The experimental data shows that E94 plays a key role in desensitization and the authors argue that this is due to the interactions of this residue with the β10-11 linker. However, the MD simulations show that these interactions happen for a small fraction, ~10%, of the time and with interaction energies comparable to those of the H98-E107-D109 cluster. It is not clear how these sparse and transient interactions can play such a critical role in desensitization. Also, if the interaction energies are of the same sign, how come one set of mutants favors desensitization and one does not?

The 10% value is the amount of time when at least a hydrogen bond forms between E94/R94 and the β10–β11 loop. It is NOT the amount of time that they form interactions, as there could be other types of non-bonded interactions such as Van der Waals interaction and Coulombic interaction. In fact, our non-bonded energy calculation clearly suggests that R94 interacts with the β10–β11 loop much more favorably than E94 (Figure 4C). The impact of E94R on β10–β11 loop is also reflected in the root-mean-square-fluctuation analysis, where the β10–β11 loop shows a reduced flexibility when R94 is present (Figure 4B).

Our central hypothesis is that PAC becomes more prone to desensitization when the desensitized conformation is stabilized. Two critical interactions are characteristic of the desensitized structure of PAC, including the association of the E94 with the β10–β11 loop, and H98 with E107/D109. Therefore, we expect mutations that alter these interactions to affect PAC channel desensitization. Based on the MD simulations, we observed the root-mean-square-fluctuation of β10–β11 loop are reduced for E94R when compared to WT (Figure 4B), suggesting that β10–β11 loop is stabilized when E94 is replaced by an arginine. The non-bonded interaction energy between E94 and the β10–β11 loop is also more negative for E94R when compared to WT (Figure 4C), another indicator of conformation stabilization. As a result, the E94R mutant favors desensitization. This is in sharp contrast with the H98R data, in which H98R interact less favorably with E107/D109 (Figure 2F, G, H, I) when compared to WT. Although the interaction energies are of the same sign, it is the difference between WT and the mutants that will ultimately determine whether a certain mutation will favor desensitization or not.

2.4. Are 600 ns sufficient to evaluate sampling of the different conformations? The MD simulations should be extended beyond the 600 ns to increase sampling, ideally, multiple independent repeats should be carried out and analyzed at different pHs.

Our MD analysis doesn’t intend to sample large conformational transitions between different functional state. Instead, our analysis focused on local dynamics which allowed us to correlate the observation with electrophysiology data. During the revision, we have extended our simulation to 1 μs for each mutant. It is worth pointing out that because PAC protein is a trimer, and we performed all the calculations across three subunits. Therefore, the effective sampling time would become 3 μs in total. Overall, the added simulation result remains the same with our initial analysis, suggesting that the sampling time is sufficient to evaluate the metrics reported in the study. We also acknowledged this limitation of our study in the discussion of the revised manuscript.

3. The underlying assumption in the interpretation of all the data is that the mutations stabilize or destabilize the desensitized conformation of the channel. However, none of the functional measurements provide direct evidence supporting this key assumption. The conclusions would be greatly strengthened if the authors could directly show that the mutations speed up or slow down the rate of recovery from the desensitized state. This should be feasible for most constructs as activation and desensitization happen at different pHs.

We agree with the reviewer that our functional data measure the degree and rate of the PAC channel entering desensitization from the activated state upon prolonged acid treatment. This is a common experimental procedure for research on desensitization/inactivation of ion channels. Following the reviewer’s suggestion, we also sought to capture the kinetics from the desensitized state to the activated state by switching from more acidic pH to less acidic pH (for example 4.0 to 5.0) or neutral pH. However, we found that such experiments are not feasible partly because the kinetics of PAC desensitization is much slower compared to other channels, such as ASIC channels (see a recent study we cited: https://elifesciences.org/articles/51111). For the mutants with strong desensitization (E94R and D91R), it’s unclear whether the currents we recorded at pH 5.0 right after pH 4.0 representing the activated state or the desensitized state at pH 5.0. In other words, we don’t know if the PAC channel transitions from the desensitized state from a lower pH back to the activated state or rather directly to the desensitized state at a higher pH. For the mutants with reduced desensitization, the current amplitude at pH 4.0 were often similar to that at pH 5.0, which makes the recovery/transition variable. We also tried to switch the acidic pH to neutral pH. We found that the PAC channels (both WT and mutants) go back to the closed state from the desensitized state in seconds as limited by our perfusion speed. These data suggest that the desensitized state of PAC is no longer maintained after switching buffer from low pH to neutral pH. In summary, it’s technically infeasible, in our opinion, to measure the rate of recovery from desensitization to activation for the slowly desensitizing PAC channel. However, our data do support the conclusion that the rates of entering desensitization from the activated state, a standard measurement of desensitization, change for various channel mutants we studied.

4. The reliability of the reported time constants of inactivation is questionable. For the reliable fitting of an exponential function, a time course at least 2-3x longer than the time constant itself must be fitted. In many cases, the presented time courses are too short to afford an exponential fit. E.g.: Figure 2B (H98R, E107R); Figure 3B (WT at pH 4.6 or 5.0, E94Q at pH 4.6 or 5.0); Figure 4E (H98R, E94R/H98R); Figure 5 (WT at pH 4.6 or 5.0, D91N at pH 4.6 or 5.0); Figure 6B (E250R).

We thank the reviewer for the helpful suggestion. The normalized current fraction (30s/max) and time constant were extracted from the same recording data, and the two parameters indeed correlated with each other very well. We agree with the reviewer that the normalized current fraction is a simpler and more reliable parameter, which is sufficient to support our conclusions. Therefore, we have now removed the time constant panels from the figures and directly added some of them (fast decay ones) in the text.

Reviewer #1 (Recommendations for the authors):1. I am concerned about the reliability of the reported time constants of inactivation. For the reliable fitting of an exponential function, a time course at least 2-3x longer than the time constant itself must be fitted. In many cases, the presented time courses are too short to afford an exponential fit. E.g.: Figure 2B (H98R, E107R); Figure 3B (WT at pH 4.6 or 5.0, E94Q at pH 4.6 or 5.0); Figure 4E (H98R, E94R/H98R); Figure 5 (WT at pH 4.6 or 5.0, D91N at pH 4.6 or 5.0); Figure 6B (E250R). The normalized surviving current fraction after 30 s is a simpler and more reliable parameter – maybe it would suffice for supporting the authors' conclusions, even if reliable time constants cannot always be extracted from the data.

We thank the reviewer for the helpful suggestion. The normalized current fraction (30s/max) and time constant were extracted from the same recording data, and the two parameters indeed correlated with each other very well. We agree with the reviewer that the normalized current fraction is a simpler and more reliable parameter, which is sufficient to support our conclusions. Therefore, we have now removed the time constant panels from the figures and directly added some of them (fast decay ones) in the text.

Reviewer #2 (Recommendations for the authors):Overall, this is a nice study with well-done experiments. Interpretations of the data seem largely reasonable, though some questions remain about exactly how to reconcile structural and functional data and the precise molecular mechanisms underlying desensitization. I have several comments that could improve the interpretability and accessibility of the study for a general audience.1. The major issue is that important regions for desensitization are implicated when the model would be much more complete if specific protonation events involved were identified or at least clearly hypothesized. Can one say definitively that additional protonation events after opening are required as indicated in the model figure? Are they on residues implicated here? Several additional mutants could be evaluated to address this: (1) E107Q, (2) D109N, and (3) E107Q/D109N to explore protonation of the acidic pocket and (4) E249Q, (5) E250Q, and (6) D251N to explore protonation of the beta10-11 linker (in addition to charge reversal mutations presented). In addition, predicted pKas in wild-type and mutant structures for the relevant acidic residues could be presented to guide possible protonation mechanisms and explain mutant effects. For example, altered desensitization in H98R could be explained by perturbed pKa of nearby E107/D109 in addition to (or instead of) reduced interaction strength between the 98-109 side chains.

We thank the reviewer for the suggestion. We generated the recommended mutants and performed new experiments (except E249Q and D251N for which the charge-reversal mutations had normal desensitization). Similar to E107R and D109R, the E107Q and D109N mutants exhibited reduced desensitization at pH 4.0 (Figure 2B, C). These results suggest that the protonation of these two residues may partially mediate the pH-dependence of PAC desensitization by weakening the association of H98 with E107/D109 in the ECD-TMD interface. The double mutant did not further decrease the current decay, consistent with the idea that they likely work through the same structural mechanism. In contrast, the E250Q mutant exhibited normal desensitization as WT PAC channel (Figure 6B, C). This suggests that the protonation of E250 does not underlie the pH-dependence of PAC desensitization.

We also carried out the pKa prediction of E107 and D109 for both WT and H98R as the reviewer suggested. The result is summarized Table 1.

As shown from the table, the H98R mutant only caused a small pKa upshift for both E107 and D109. At the pH range where desensitization is observed (pH 4.0), D109 will remain unprotonated, whereas E107 will be protonated. Therefore, we don’t think altered pKa of E107/D109 is a major contributor for the reduced desensitization phenotype of H98R.

2. Related to 1, the model figure 7 would benefit greatly if protonation states of residues (or groups of residues) are indicated, especially as previous work has implicated residues in different processes.

Although our study clearly identified a set of residues that play a critical role in PAC channel desensitization, we believe these residues regulate this process through complex mechanisms. While the new data presented in this revision suggests that protonation of E107 and D109 can account for the pH-dependent PAC desensitization to some degree, it is unlikely the sole mechanism. Our Figure 7 intends to show that polar interactions encoded in the ECD-TMD interface are important for PAC desensitization, not just through the protonation events of these residues per se. Therefore, we have revised the cartoon in Figure 7 to make less emphasis on the protonation events. We hope that this will cause less confusion and convey a better summary of our study.

3. I do not understand the explanation provided for the difference between the properties of endogenous PAC and overexpressed PAC in a PAC -/- cell line, which is substantial. Why should overexpression (resulting in ~4x current density) results in a higher degree and faster rate of desensitization? This is an interesting difference to point out even if the reason for the difference is unclear.

Yes, it’s an interesting difference. We don’t know the reason, but it suggests that a factor/mechanism limiting PAC desensitization in the endogenous setting is overwhelmed by the increased number of overexpressing channels. We added a sentence in the result: Interestingly, its degree and kinetics appeared to be greater and faster compared to the endogenous currents. This difference suggests a potential endogenous factor/mechanism limiting PAC desensitization that is overwhelmed by the large number of overexpressing channels.

4. A brief discussion further connecting the structure papers with this work would help with context. How do the proportions of open, closed, and desensitized structural states at high and low pH values (refs 17,18, Deng et al. Sci. Adv. 2021) match the expected distribution based on the recordings here? Can the authors explain discrepancies in different conditions (e.g. detergent vs nanodisc)?

We are aware of the populations of PAC in various conformations from Deng et al. Because our electrophysiology experiment can only measure macroscopic current, it does not carry information about the proportions of the channel in different conformations, especially the resting and the desensitized states that are both observed at pH 4.5 from structural analysis. The exact reason why the desensitized state becomes more prevalent in nanodisc than detergent is also unclear in Deng et al. One possible explanation could be that specific types of lipid may play an important role in favoring the desensitized state. Indeed, we found that PIP2 regulates PAC channel activity in an unconventional mechanism in another study (https://doi.org/10.1101/2022.10.06.511171). We have added this conjecture in our discussion of the revised manuscript.

5. Line 212 appears to include a typo. Residue107-249 intersubunit and 250-297 intrasubunit interactions are observed in PDB 7SQH.

We updated the text to mention the E107-E249 intrasubunit interaction.

6. This sentence may be misworded. "Specifically, mutations predicted to stabilize the low-pH non-conducting structure (H98R, E107R, D109R, and E250R) greatly reduced channel desensitization; on the other hand, those predicted to destabilize it (E94R and D91R) produced the opposite effect (Figure 7)." H98R, E107R, D109R, and E250R should destabilize the desensitized state to reduce desensitization.

Yes, we revised accordingly.

7. Figure 7 legend: does the model depict one or two subunits?

Two subunits, we now added it to the figure legend.

Reviewer #3 (Recommendations for the authors):Suggestions:I think it is critical to directly show that the mutations speed up or slow down the recovery from desensitization. This should be feasible for most constructs as activation and desensitization happen at different pHs.

We agree with the reviewer that our functional data measure the degree and rate of the PAC channel entering desensitization from the activated state upon prolonged acid treatment. This is a common experimental procedure for research on desensitization/inactivation of ion channels. Following the reviewer’s suggestion, we also sought to capture the kinetics from the desensitized state to the activated state by switching from more acidic pH to less acidic pH (for example 4.0 to 5.0). However, we found that such experiments are not feasible partly because the kinetics of PAC desensitization is much slower compared to other channels, such as ASIC channels (see https://elifesciences.org/articles/51111). For the mutants with strong desensitization (E94R and D91R), it’s unclear whether the currents we recorded at pH 5.0 right after pH 4.0 representing the activated state or the desensitized state at pH 5.0. In other words, we don’t know if the PAC channel transitions from the desensitized state from a lower pH back to the activated state or rather directly to the desensitized state at a higher pH. For the mutants with reduced desensitization, the current amplitude at pH 4.0 were often similar to that at pH 5.0, which makes the recovery/transition variable. We also tried to switch the acidic pH to neutral pH. We found that the PAC channels (both WT and mutants) go back to the closed state from the desensitized state in seconds as limited by our perfusion speed. These data suggest that the desensitized state of PAC is no longer maintained after switching buffer from low pH to neutral pH. In summary, it’s technically infeasible, in our opinion, to measure the rate of recovery from de-sensitization to activation for the slowly desensitizing PAC channel. However, our data do support the conclusion that the rates of entering desensitization from the activated state, a standard measurement of desensitization, changed for various channel mutants we studied.

The MD simulations should be extended beyond the 600 ns to increase sampling, ideally, multiple independent repeats should be carried out and analyzed at different pHs.The effects of protonation choices on different should be evaluated.

Our individual simulation is 200 ns in length. In our initial manuscript, we carried out three independent simulations for each mutant, with a total amount of sampling time for 600 ns. In this revision, we extended the simulations to 1 μs. The quantification remains consistent with our initial result, suggesting the convergence of our simulation. We focused on pH 4.0 as the desensitization is most pronounced at this condition. It is worth mentioning that none of the residues of interest (H98, E107, D109, E250, E94, and D91) have a predicted pKa between 4.0-4.5. Therefore, the predominant protonation status of these residues will not change at the relevant pH range. We also acknowledge the limitations of our strategy by just simulating the predominant form of the titratable sites, as the protonation of these residues should be a dynamic process in reality. This caveat is now mentioned in the discussion of our revised manuscript.

The inconsistencies between electrophysiological results and MD simulations should be addressed and, if it is not possible to resolve them, then they should be acknowledged.

We now acknowledge part of the result where limitation is present.

- I do not see a reduction of interaction energy and of distance with E107 in H98 vs 98R (Figure 2), as claimed in the text (pg. 7).

The reduction of interaction energy between E107/R98 is relatively small, but one should be able to appreciate that the non-bonded energy between H98/E107 samples more between -20 to -40 kJ/mol range than R98/E107 (Figure 2H). We have now revised the text to better describe this result.